# Relationship between Satisfaction Scores and Racial/Ethnic and Sex Concordance in Primary Care

**DOI:** 10.3390/healthcare11162276

**Published:** 2023-08-12

**Authors:** Rebekah J. Walker, Aprill Z. Dawson, Abigail Thorgerson, Jennifer A. Campbell, Sara Engel, Mandy Kastner, Leonard E. Egede

**Affiliations:** 1Division of General Internal Medicine, Department of Medicine, Medical College of Wisconsin, Milwaukee, WI 53226, USA; rebwalker@mcw.edu (R.J.W.); adawson@mcw.edu (A.Z.D.); jacampbell@mcw.edu (J.A.C.); 2Center for Advancing Population Science, Medical College of Wisconsin, Milwaukee, WI 53226, USA; aathorgerson@mcw.edu; 3Department of Emergency Medicine, Medical College of Wisconsin, Milwaukee, WI 53226, USA; saraengel@mcw.edu; 4Department of Medicine, Medical College of Wisconsin, Milwaukee, WI 53226, USA; mkastner@mcw.edu

**Keywords:** satisfaction with care, racial/ethnic, sex, concordance, primary care

## Abstract

Racial/ethnic and sex concordance between patients and providers has been suggested as an important consideration in improving satisfaction and increasing health equity. We aimed to guide local efforts by understanding the relationship between satisfaction with care and patient–provider racial/ethnic and sex concordance within our academic medical center’s primary care clinic. Methods: Satisfaction data for encounters from August 2016 to August 2019 were matched to data from the medical record for patient demographics and comorbidities. Data on 33 providers were also obtained, and racial/ethnic and sex concordance between patients and providers was determined for each of the 3672 unique encounters. The primary outcome was top-box scoring on the CGCAHPS overall satisfaction scale (0–8 vs. 9–10). Generalized mixed-effects logistic regression, including provider- and patient-level factors as fixed effects and a random intercept effect for providers, were used to determine whether concordance had an independent relationship with satisfaction. Results: 89.0% of the NHW-concordant pairs and 90.4% of the Minority Race/Ethnicity-concordant pairs indicated satisfaction, while 90.1% of the male-concordant and 85.1% of the female-concordant pairs indicated satisfaction. When fully adjusted, the female-concordant (OR = 0.58, 95% CI 0.35–0.94) and male-discordant (OR = 0.68, 95% CI 0.51–0.91) pairs reported significantly lower top-box satisfaction compared to the male-concordant pairs. Significant differences did not exist in racial/ethnic concordance. Conclusions: In this sample, differences in sex concordance were noted; however, patient- and provider-level factors may be more influential in driving patient satisfaction than race/ethnicity in this health system.

## 1. Introduction

Patient satisfaction has become a staple metric utilized across the healthcare industry for patients and insurers to measure individual provider and overall health system performance [1,2]. Despite the complex nature of the healthcare system and factors that may differ across outpatient and inpatient settings, one of the most consistent drivers of patient satisfaction noted in the literature is the patient–provider relationship [2,3]. While initial studies focused on differences in patient or provider demographics, concordance on demographics between providers and patients was later highlighted as a factor in patients’ satisfaction ratings [1,2].

Concordance can be defined as a state of agreement or harmony and is generally investigated across domains such as sex, social class, age, ethnicity, race, language, sexual orientation, beliefs about roles, beliefs about health and illness, values, and actual healthcare decisions [1]. Investigations into how concordance can improve care suggests that it can influence process variables such as provider–patient communication, patient knowledge/understanding, patient adherence, and the appropriateness of care [1]. Studies suggest patients who prefer concordant physicians do so primarily because of concerns regarding language and empathetic treatment [1,4]. For example, an evaluation of patient visits found that race-concordant pairs had longer patient visits and patients reported greater ease in discussing problems [4].

Prior research on the association between patient–provider racial/ethnic concordance and outcomes is mixed [5]. Racial/ethnic concordance has been found to be associated with healthcare utilization, specifically with race-concordant patient–provider dyads being associated with lower odds of not utilizing needed health services, lower odds of delayed care seeking, and greater odds of completing doctor visits in the past year [6,7]. In addition, patient–provider concordance has been found to be associated with higher ratings of patient satisfaction with care and improved patient–provider communication [1]. While racial/ethnic concordance is known to be associated with increased patient satisfaction and an increased likelihood of physicians receiving the maximum satisfaction score, patient-level factors such as health status, functional limitations, insurance, and age are additional key drivers of patient satisfaction with care [8,9]. Interestingly, one study found that racial/ethnic concordance was not associated with the health outcome of blood pressure control, nor was it associated with patient communication, a factor known to be associated with patient satisfaction [10,11]. In addition, race concordance was not associated with increases in preventive screening practices using nationally representative data [12].

Similarly, the results of prior studies on sex concordance are mixed, though the majority note an important impact on outcomes [8,13,14,15,16]. A meta-analysis revealed a stronger relationship between sex concordance and clinical outcomes than it did on the assessment of communication, trust, or patient assessment of physician performance [14]. In particular, female concordance was more beneficial than male concordance in studies that assessed survival rates following acute myocardial infections, appropriate antibiotic prescribing patterns and treatment for diabetes [14]. Male concordance was associated with improved compliance with weight-related counseling in obese patients but had limited influence on other preventive care practices [14]. While women with sex-concordant providers were more compliant with mammography, the impact was not clinically meaningful [15]. In a study conducted across 10 clinics, perceived personal and cultural similarity based on values, beliefs, and use of patient-centered communication was more influential in explaining patient satisfaction than concordance in sex or race [13].

Given the mixed evidence, and the importance of local contexts in understanding patient satisfaction, the objective of this study was to examine the relationship between provider–patient concordance in sex and race/ethnicity, and patient satisfaction with care in an ambulatory care setting. As a unique contribution to the field, we used generalized mixed-effects models to account for the clustering of patients within providers when assessing overall provider satisfaction ratings. 

## 2. Materials and Methods

### 2.1. Study Design, Data Source, and Study Population

This study was a cross-sectional analysis using data collected from our institutional Clinician and Group Consumer Assessment of Healthcare Providers and Systems (CGCAHPS) surveys for responses reported on visits completed between August 2017 and August 2019.

The Froedtert/Medical College of Wisconsin (MCW) health system is one of the primary health systems in the Southeast Wisconsin region of the United States. The primary hospital is located in Milwaukee County, with additional hospitals located in the surrounding counties. Health centers and clinics are located throughout the region. Milwaukee County is racially and ethnically diverse, with its percentage of African American/Black residents higher than that in the average US population and its percentage of Hispanic/Latino residents similar to that of the average US population. The surrounding counties have lower percentages of minority populations compared to Milwaukee County; however, they are more similar to the average US demographic makeup compared to other counties in Wisconsin. The sex and age demographics for the region are similar to those of the average US population.

Only providers that completed patient visits as part of the General Internal Medicine Division were used. Froedtert/MCW offers the only academic medical center primary care clinic in the region. The clinic consists of attending physicians, Advanced Practice Providers (APPs), and residents, with a clinic work week for full time providers consisting of eight clinic half-days. Clinic visits where attending physicians oversaw residents were not included in this analysis.

Information on patients was abstracted from their electronic health records and information on providers was obtained from Division and Human Resources. The CGCAHPS surveys included in this study contained 33 providers across 3672 unique patient encounters.

### 2.2. Primary Outcome

The primary outcome was the overall provider satisfaction rating, which was recorded on a scale of 0–10. Top-box dichotomization was used, where scores of 0 to 8 were considered ‘unsatisfactory’ and scores 9 to 10 were considered ‘satisfactory’.

### 2.3. Independent Variable

Sex and race/ethnicity concordance between patient and provider was created based on variables abstracted from the medical record for patients and provided by human resources for providers. Sex was reported in each case as male or female. Using the sex noted for each individual, a concordance variable was created with four levels. Female concordance was defined as both the patient and provider being female. Male concordance was defined as both the patient and provider being male. Female-discordant pairs included a female provider and male patient. Male-discordant pairs included a male provider and female patient.

Race/ethnicity was reported as White, African American/Black, Hispanic/Latino, and other race. Due to sample size considerations, the categories were collapsed into Non-Hispanic White (NHW) and Minority Race/Ethnicity. A concordance variable was then created with four levels. NHW concordance was defined as both the patient and provider being NHW. Minority concordance was defined as both the patient and provider being Non-White. NHW-discordant pairs included an NHW provider and a Non-White patient. Minority-discordant pairs included a Non-White provider and an NHW patient.

### 2.4. Covariates

Both patient and provider variables were included as covariates. Provider-level factors included age (continuous), sex (dichotomous), race/ethnicity (Non-Hispanic White or Minority Race/Ethnicity), provider type (MD or Advanced Practice Provider (APP)), years since medical school/medical training (continuous), section (primary care or peri-operative medicine), percent clinical effort (continuous), and faculty rank (Assistant, Associate Professor, or Full Professor). Patient-level factors included age (continuous), sex (dichotomous), race/ethnicity (Non-Hispanic White or Minority Race/Ethnicity), insurance status (Managed Care, Medicaid, Medicare, Self-pay), and epic risk score (continuous measure with higher scores indicating greater clinical complexity).

### 2.5. Statistical Analysis

All analyses were conducted using STATA version 15. A *p*-value less than 0.05 was considered statistically significant. Summary statistics of the patient and provider variables were summarized separately. Numeric variables were presented as means and standard deviation, while categorial variables were presented as counts and percentages. Generalized mixed-effects logistic-regression was employed to investigate associations between concordance and top-box satisfaction with the clinic visit while accounting for the nesting of patients within providers. Two unadjusted models (one with race/ethnicity concordance as the primary outcome and the second with sex concordance as the primary outcome) were run with random effects on the providers due to patients being nested within providers. The second set of models was adjusted with for provider factors (age, sex, race/ethnicity, provider type, years since medical school, section, percent clinical effort, and faculty rank). The third set of models was adjusted for patient factors (age, sex, race/ethnicity, insurance status, education, and epic risk score). Finally, the fourth set of models was adjusted for both patient and provider factors. Patients who had multiple encounters with the same provider were treated as one observation for each patient with the mean satisfaction score across visits used as the outcome. Those who had multiple encounters with different providers were treated as separate observations. Hence, each observation had a unique patient/provider combination in the data.

## 3. Results

The average number of patients clustered within providers was 138, with a range of 4 to 506 patients. The intraclass correlation (ICC) was 15.1% (95% CI: 8.6–25.0). ICC shows the amount of variance explained by clustering at the provider level. This ICC shows significant clustering of top-box satisfaction scores within providers, therefore justifying the use of mixed-effects logistic regression models.

Table 1 provides a summary of the patient and provider characteristics. Most patients and providers were female (66.5% and 69.7%, respectively). A total of 21.0% of patients and 18.2% of providers were of a Non-White minority race.

Table 2 provides the results of the models with sex concordance as the primary independent variable. In the unadjusted mixed-effects logistic regression model, female-concordant (OR = 0.35, 95% CI 0.18–0.68), female-discordant (OR = 0.45, 95% CI 0.22–0.90), and male-discordant (OR = 0.65, 95% CI 0.51–0.83) pairs had significantly lower odds of indicating top-box satisfaction scores compared to male-concordant pairs. When fully adjusted, only the female-concordant (OR = 0.58, 95% CI 0.35–0.94) and male-discordant (OR = 0.68, 95% CI 0.51–0.91) pairs maintained significantly lower top-box satisfaction compared to male-concordant pairs.

Table 3 provides the results of the models with racial/ethnic concordance as their primary independent variable. In the unadjusted mixed-effects logistic regression model, Non-Hispanic White concordance (OR = 1.51, 95% CI 1.09–2.10) had significantly higher odds of indicating top-box satisfaction scores compared to Non-Hispanic White discordant pairs. However, after adjustment, there were no statistically significant differences between racial/ethnic concordance groups.

## 4. Discussion

This study conducted in an academic medical center in Midwest US found that male concordance was a stronger driver of top-box scores compared to female concordance or discordance in sex. Race/ethnicity concordance was not a significant driver of top-box scores after adjustment for patient and provider factors, and accounting for the clustering of patients within providers. Although replication with larger samples is necessary, these findings are important given the broad definition of both the patient- and provider-level covariates used to identify whether sex or racial/ethnic concordance was independently associated with patient satisfaction. Based on the results of this study, individual patient- and provider-level factors may be more influential in driving patient satisfaction than patient–provider race/ethnicity concordance in this health system.

This study adds to the literature by investigating differences in the impact of concordance on patient satisfaction, while accounting for patient- and provider-levels factors and accounting for clustering in the data. Existing evidence on the roles of sex and race concordance on patient satisfaction has been mixed [8,9,17]. For example, a recent study using a national sample of adults found that patient/provider racial/ethnic concordance was modestly associated with increased satisfaction in care and that other drivers, such as functional limitation, age, and insurance, were more significantly associated with satisfaction [9], whereas a study from a northeastern health system found that racial/ethnic concordance was significantly associated with higher patient satisfaction [8]. Our results found that racial/ethnic concordance was not significantly associated with patient satisfaction after accounting for important confounders. Prior work on sex concordance found that overall satisfaction was not impacted by provider sex concordance when receiving care in the ED [18]. In that study, women preferred sex-concordant providers, but had lower satisfaction [18]. Similarly, in a sample of Medicaid patients, Prasad and colleagues found that sex concordance had no significant relationship with patient satisfaction [19]. In our sample, sex concordance was significantly associated with patient satisfaction and male concordance resulted in the highest satisfaction rating.

There are several implications of this work for everyday practice. First, as patient satisfaction is a key performance metric, healthcare systems should continue working to understand how to emphasize patient-centered approaches across all providers, regardless of concordance. Local context should be considered across health systems to understand patient preferences in care settings, as one-size-fits-all processes when it comes to the patient–provider relationship may not be appropriate for all settings. Quality improvement and research should focus on understanding patient priorities and the role of healthcare system factors external to the provider in patient satisfaction scores. Our team is conducting a mixed-methods study to collect qualitative data that can inform local efforts focused on improving patient satisfaction. Secondly, this work highlights the importance of embedding health service researchers into health systems to support new methodological approaches to important clinical questions. For example, in this analysis, unadjusted results could have been misleading, requiring the thoughtful selection of covariates and sophisticated models that accounted for the clustered nature of the data. By integrating clinical and research perspectives into health systems work, more comprehensive and informative analyses could be conducted to provide clinically meaningful results.

While this analysis was strengthened by accounting for both patient- and provider-level factors in understanding the association with patient satisfaction with care, the study has limitations worth noting. First, the study findings are not generalizable to other health systems, as data from the One Health system in the Midwestern United States were used and findings may vary across US regions. Second, structural factors that may influence patient satisfaction, including experiences with discrimination or racism and implicit bias in healthcare, are not accounted for within this analysis. Efforts to identify how to capture this information and investigate the influence on patient satisfaction will be important for future work. Third, the primary comparisons in this study were sex and race/ethnicity. The categorization used, which was based on the data available, does not necessarily capture the experiences and viewpoints of all genders or racial/ethnic groups. Studies with additional information on genders beyond traditional male and female, and with larger sample sizes, allowing for concordance across multiple minority groups, should be conducted in the future. Finally, while top-box scores were used to mimic national and institutional reporting guidelines on patient satisfaction, this minimizes variability in the data and does not adequately use the full range of scoring that could better explain differences between providers.

## 5. Conclusions

In conclusion, the satisfaction scores in an academic medical center’s primary care clinic were higher for male-concordant pairs and male patients with female providers compared to female-concordant pairs after adjusting for patient- and provider-level factors. Significant differences did not exist in racial/ethnic concordance. These findings highlight the importance of patient- and provider-level factors in driving patient satisfaction beyond simple summaries such as concordance, and the importance of conducting well-designed analyses that help identify the factors that influence patient satisfaction.

## Figures and Tables

**Table 1 healthcare-11-02276-t001:** Patient and provider characteristics.

**Provider Factors**	**(n = 33)**
Women	69.7%
Minority Race/Ethnicity	18.2%
Age	48.1 ± 11.0
Years since graduation	18.0 ± 11.1
Percent clinical effort	0.7 ± 0.3
Faculty rank	
Assistant Professor	36.4%
Associate Professor	36.4%
Full Professor	27.2%
**Patient Factors**	**(n = 3672)**
Women	66.5%
Minority Race/Ethnicity	21.0%
Age	59.6 ± 15.8
Epic risk score	1.5 ± 1.6
Insurance	
Managed Care	33.4%
Medicaid	3.7%
Medicare	47.2%
Self-pay	14.4%

**Table 2 healthcare-11-02276-t002:** Generalized mixed-effects logistic regression on top-box satisfaction for sex concordance.

	Unadjusted OR (95% CI)	Adjusted OR (95% CI) ^1^
Male Concordance	1.00	1.00
Female Concordance	0.35 (0.18–0.68) *	0.58 (0.35–0.94) *
Male Discordance	0.65 (0.51–0.83) *	0.68 (0.51–0.91) *
Female Discordance	0.45 (0.22–0.90) *	0.72 (0.40–1.28)

** p*-value < 0.05. ^1^ adjusted for patient- and provider-level covariates.

**Table 3 healthcare-11-02276-t003:** Generalized mixed-effects logistic regression on top-box satisfaction for race/ethnicity concordance.

	Unadjusted OR (95% CI)	Adjusted OR (95% CI) ^1^
Non-Hispanic White Discordance	1.00	1.00
Non-Hispanic White Concordance	1.51 (1.09–2.10) *	1.23 (0.92–1.66)
Minority Concordance	1.91 (0.59–6.21)	1.97 (0.60–6.43)
Minority Discordance	0.89 (0.53–1.50)	0.82 (0.44–1.52)

* *p*-value < 0.05. ^1^ adjusted for patient- and provider-level covariates.

## Data Availability

Data is unavailable due to privacy restrictions.

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
