# Peer review of "Relationship between Satisfaction Scores and Racial/Ethnic and Sex Concordance in Primary Care"

_healthcare, 2023, doi:10.3390/healthcare11162276_

Round 1

Reviewer 1 Report

Dear Authors,

The presented study tackles an issue of relationship between satisfaction scores and racial/ethnic and sex concordance in Primary Care. I have read the article with a great interest. The study was conducted reliably with appropriate selection of tests. Overall, I think that this article should be published, however some issues require complementary information:

1.       Please change references according to journal’s guidelines.

2.       Please add ethics committee approval number.

3.       I suggest adding what are the implications for everyday practice from your study.

Author Response

We thank the reviewer for their comments. Revisions have been made to the manuscript to 1) update references to journal guidelines per notes within the manuscript from the editor, 2) comment on ethics approval within the manuscript, and 3) clearly identifying the third paragraph of the discussion is intended to comment on implications for everyday practice - namely the importance of understanding local context in patient satisfaction and the importance of incorporating health services researchers into health systems support services.

Reviewer 2 Report

Dear authors,

It is a pleasure to review your article titled "Relationship between Satisfaction Scores and Racial/Ethnic and Sex Concordance in Primary Care." I have carefully evaluated your work and I am pleased to share with you my comments and suggestions to improve your article.

Firstly, I would like to congratulate the team of authors on the approach and methodology employed in your research. In my opinion, the topic is relevant and interesting for both healthcare professionals and healthcare users.

However, I have also identified some areas that could be improved to strengthen your article. I will now detail the aspects that I consider necessary to address:

  • Materials and Methods: I recommend stating the study design at the beginning of this section, preferably in a subsection.

  • Discussion: I believe the authors' results are clear. However, it would be beneficial to expand the discussion and delve deeper into them. This is evident from the fact that approximately half of this section is dedicated to the implications and limitations of the study. For example, "Whereas a study from a northeastern health system found that racial/ethnic concordance was significantly associated with higher patient satisfaction.8 Our results found that racial/ethnic concordance was not significantly associated with patient satisfaction after accounting for important confounders" (lines 189-192). Why might there be this difference in the results? It would be interesting for the scientific community to discuss and justify the differences between studies. Additionally, to enhance readability, it is recommended to differentiate the limitations of the study with a subsection.

  • References: I have noticed that only 19 references have been included. I urge you to review and add more citations to enrich the discussion section.

Thank you for your attention to these comments, and I encourage you to address these points in your article revision. I look forward to seeing a revised version that takes these suggestions and improvements into account.

Once again, I appreciate your contribution and congratulate you on your work. I am confident that with the appropriate revisions, your article will be even stronger and more valuable to the scientific community. Please feel free to reach out if you have any further questions.

Best regards.

Author Response

We thank the reviewer for their feedback and comments on the approach and methodology used within the study. We addressed points raised by the reviewer by updating the initial section of the Methods to state Study Design in the sub-section heading and state that this was a cross-sectional analysis. While we appreciate the interest in commenting on why our results may differ from prior studies, we do not have data that can provide insight of that kind currently. We added a statement in the third paragraph of the discussion to note that additional work is ongoing to collect qualitative data that will help the team understand differences between these findings an prior work in other health systems. 

Reviewer 3 Report

The Topic looks exciting and timely.

There are a few concerns that must be addressed by the authors:

            1.      Generally, PRISMA guidelines are used for Systematic Review.

The authors must include the PRISMA process for better understanding.

2.      What conditions are used to include a case for the current study?

3.       In 195-197 Similarly, in a sample of Medicaid patients…highest satisfaction rating.

Authors must discuss why their findings are different from the literature. What are the implications of the results?

4.      The discussion section fails to portray the value of the study as it could. This could be done by starting with a description of the contribution of the findings. Thereafter the notion of the association between gratification scores and personality characteristics such as Racial/Ethnic and Sex concordance in Primary Care.

5.      The authors should discuss the limitation of the paper

6.      The authors should discuss the “Implications (Theoretical and Practical)” of the paper

7. The quality of the contribution of this work can be assessed once there is a better understanding of the measures used. At face value, the findings offer significant and exciting relationships to help understand the association between gratification scores and personality characteristics, but the communication of the findings concerning theory should be improved.

Moderate English editing could make the content more suitable.

Author Response

We thank the reviewer for feedback. As this study was not a systematic review we did not follow PRISMA guidelines for reporting. We also outline both implications and a number of limitations in our discussion, which are requested by the reviewer. As a result, we believe this review may have been completed on a different manuscript as the points raised are highly relevant, but do not match to this manuscript.

Reviewer 4 Report

The theme of the article is interesting and topical.
The data analysis is quite rigorous and scientifically sound.

However, I believe that in the current form the paper still comes in the form of a draft.

Bibliographic references are few and in many cases outdated. The authors should make more use of the relevant literature on the topic so as to fully engage in the debate.

The conclusions represent a summary of the work, and offer no food for thought.

It should contain clear future policy or research directions from the findings.

Likewise, the authors do not describe at all the social, political and cultural context of the area in which the research was conducted. I consider this to be a serious shortcoming. Readers might appreciate the work more in context. 

Author Response

We thank the reviewer for feedback on the manuscript. We aimed to focus the literature review on the most relevant articles summarizing patient satisfaction specific to provider-patient concordance in race/ethnicity and sex. As a result, while we agree there are many more possible citations on satisfaction, those included in the manuscript are highly relevant for the topic of concordance investigated in this study. We have added comments on implications and noted ongoing work to collection qualitative data and guide improvement efforts. As this was quality improvement work and not intended to inform policy at a broader scale we chose not to comment on that in this manuscript to ensure our discussion was focused within the context of the data analyzed.

Round 2

Reviewer 4 Report

The new version of the paper is quite similar to the previous one. 

I continue to see few and outdated references. 

The conclusions are not useful for readers.

There is not a description of the social, political and cultural context of the area in which the research was conducted. 

For all these reasons, I reject the publication of the paper.

Author Response

  1. I continue to see few and outdated references. 

Response: References selected for inclusion were chosen based on relevance to the specific research question under investigation, as opposed to a systematic coverage of the topic of concordance. In addition, a number of systematic reviews have been conducted, which offer both coverage of prior work and summaries of gaps in the field. We included these articles instead of citing individual papers included in each review. We chose to include seminal papers in the field of patient-provider concordance, which do have older dates of publication ranging from 1997 through 2004, because we felt it was important to ground readers in how concordance has been defined and investigated in initial work on the topic. We then included articles published between 2008-2013 to summarize how to topic has been investigated in the past and the body of work that exists to date. Newer articles published between 2018-2022 provide additional insight published more recently, particularly in terms of recent work on race/ethnicity concordance.

  1. The conclusions are not useful for readers.

Response: We aimed to ensure conclusions did not reach outside the scope of this single site study, and highlighted the primary discussion points noted as relevant by the authors or other reviewers to include.

  1. There is not a description of the social, political and cultural context of the area in which the research was conducted. 

Response: Thank you for this point. We added detail regarding the area in which the health system used for this analysis is located to provide some context for the findings.